# Short Communication: IPC *Salix* Cultivar Database Proof-of-Concept

**Patrick N. McGovern** [1,*], **Yulia A. Kuzovkina** [2] **and Raju Y. Soolanayakanahally** [3]

1    Private Nurseryman and Middleware Administrator, Grand Rapids, MI 49525, USA
2    Department of Plant Science, University of Connecticut, Storrs, CT 06269, USA; jkuzovkina@uconn.edu
3    Indian Head Research Farm, Agriculture and Agri-Food Canada, Indian Head, SK S0G 2K0, Canada; raju.soolanayakanahally@canada.ca
*    Correspondence: pmcgover@gmail.com

**Abstract:** A variety of *Salix* L. (Willow) tree and shrub cultivars provide resources for significant commercial markets such as bioenergy, environmental applications, basket manufacturing, and ornamental selections. The International Poplar Commission of the Food and Agriculture Organization (IPC FAO) has maintained the Checklist for Cultivars of *Salix* L. (Willow) since 2015 and now lists 968 epithet records in a Microsoft Excel spreadsheet format. This Proof-of-Concept (POC) investigates using an SQL database to store existing IPC *Salix* cultivar information and provide users with a format to compare and submit new *Salix* cultivar entries. The original IPC data were divided into three separate tables: Epithet, Species, and Family. Then, the data were viewed from three different model perspectives: the original *Salix* IPC spreadsheet data, the Canadian (PWCC), and the Open4st database. Requirements for this process need to balance database integrity rules with the ease of adding new *Salix* cultivar entries. An integrated approach from all three models proposed three tables: Epithet, Family, and Pedigree. The Epithet and Family tables also included Species data with a reference to a website link for accepted species names and details. The integrated process provides a more robust method to store and report data, but would require dedicated IT personnel to implement and maintain long-term. A potential use case scenario could involve users submitting their Checklist entries to the Salix administrator for review; the entries are then entered into a test environment by IT resources for final review and promotion to a production online environment. Perhaps the most beneficial outcome of this study is the investigation of various strategies and standards for Epithet and Family recording processes, which may benefit the entire *Populus* and *Salix* communities.

**Keywords:** proof-of-concept: use case; spreadsheet; CSV file; SQL; database; data integrity; GitHub; Linux

## 1. Introduction

Tree breeding projects are expensive, span multiple years, and may require data from historical tree generations. These are worthy justifications to use a robust cultivar process to track epithets, families, and related details. Accurate cultivar details could provide supportive data for plant patent applications. Pedigree data from multi-generational breeding may avoid inbreeding mistakes and save time in future years. Epithet and family relationships can also be associated with nursery, field trials, and statistical views.

An online Google search for the terms "tree cultivar database" returned over one million results spanning a variety of horticultural species and topics such as urban trees, an avocado variety database, and arboretum collections. Another online Google search for "botanical database schema" returned over 500,000 results also spanning diverse subjects that addressed specific organizational needs and unrelated topics. These wide-ranging results may help explain why botanical organizations design and create their own plant databases for their specific user and technical requirements.

The venerable spreadsheet computer application is often the tool of choice for quickly storing and analyzing a variety of data. College courses have long included spreadsheet training, allowing for the widespread acceptance of spreadsheets for home, business, and scientific data. There are a variety of spreadsheet applications with proprietary file formats. However, the comma-separated (CSV) file format can be exported from most spreadsheets using a comma or other character as a field separator. These CSV text files provide a somewhat universal format to exchange data files without proprietary dependencies. Nevertheless, the ubiquitous nature of spreadsheets presents risks in terms of data integrity by not enforcing data accuracy, and in terms of the challenge of managing many different files with related data over time.

### 1.1. Characteristics of the Original IPC Salix Spreadsheet Data

In 2013, the International Poplar Commission of the Food and Agriculture Organization (IPC FAO) was appointed as the International Cultivar Registration Authority (ICRA) for willows. The first edition of the Checklist for Cultivars of *Salix* L. (Willow) [1,2] was compiled in 2015 in Microsoft Word format to promote a standardized registration process for new cultivar epithets. Eight hundred and fifty-four cultivar epithets with accompanying information were included in the first edition of the Checklist. Since then, more epithets have been added to the Checklist, which had grown to 968 records by 2020 when it was converted to a spreadsheet in Microsoft Excel format. Some duplications of epithets were revealed, making it difficult to discern unique epithet, species, and family names. There are 27 epithet names that are duplicated multiple times, including 'Pendula' (11 times) and 'Pyramidalis' (3 times). The original *Salix* spreadsheet contains 39 columns providing a flat one-line view of each epithet that does not enforce relationships between parents and other related epithets. Epithet records are encapsulated with opening and closing single quote characters [3] (e.g., 'Abbey's Harrison'). There are seven trademark names in the epithet column suffixed with the trademark sign (™) and one with the registered trademark symbol (®).

The original species field data is a mix of *Salix* species and hybrid names having 91 unique name combinations. There were 21 hybrid names that included the Punycode "×" multiplication character denoting a hybrid name. A number of the species name records were suffixed with spaces and single quote characters. There were 25 epithets with null (empty) species values and 236 with "S." representing an unknown *Salix* species name. The original *Salix* spreadsheet lists 122 records with family associations by listing the parents in the "mother" or "father" columns. The parents were a combination of species, hybrid names, and cultivars marked with single quote characters.

It should be noted that the compilation of the Checklist for Cultivars of *Salix* L. (Willow) was the first attempt to assemble scattered records from existing references, and not through direct communication with the cultivar developer and breeders. Therefore, the Checklist records lack the standard details present in the other databases investigated in this study. For example, there were limited data on pedigree and parents of most hybrid cultivars that were not identified at the clonal level. Also, there were no seedlot records or experimental trials.

### 1.2. Characteristics of the Canadian Database Model

The Canadian *Populus* and *Salix* Clone Directory [4] was produced by the Poplar and Willow Council of Canada (PWCC). The PWCC is a non-profit organization established in 1977 for wise use, conservation, and sustainable production of poplar and willow genetics resources. It stores Canadian germplasm data for clones, pollen, seedlot, and progeny. The original database was in a hard copy state since 1986, then transferred to electronic format with approximately 1000 new entries contributed by forest companies, governments, and private industry, for a total of over 6600 entries. Between 2015 and 2018, the database was converted to an online Microsoft Access database format and now contains over 26,419 records searchable by 25 column headings.

The established nature, size, and evolution of the Canadian database could provide a basis for standards for similar databases. Some Canadian database columns with a summary of their descriptions [5] and noteworthy database observations are:

- **ID:** A unique number for each entry. This column may allow duplicate "Name" column entries without conflict.
- **Name:** The family population name or a clone number or name.
- **Family:** Population entries display seedlot number. Clone entries display family numbers if known. This column data may associate with other family columns such as "Family comments", Male and Female parent columns, Male and Female Clone, and "Year Bred" columns.
- **Sex:** Four single-character designations: M (Male), F (Female), U (Unknown), or B (Both).
- **Genus/Material Type:** Five two-character designations for the genus or material type.
- **Source Type:** Multi-character categories describing source types including sib types, cuttings, wild, or NA (Not Available/Not Applicable).
- **Category:** Multi-character classifications of *Populus* or *Salix* germplasm including sib types, cuttings, wild, native, or NA (Not Available/Not Applicable).
- **Female and Male Parent Columns:** The female or male species name with parentheses used to distinguish between parents in three-species clones.
- **Female and Male Parent Clone Columns:** The female or male clone name or number used to produce the clone. Parentheses are used to distinguish between parents in three-species clones.
- **Year Bred:** The four-digit year when the breeding or collection took place or NA (Not Available/Not Applicable).
- **Year Selected:** The four-digit year when the material was selected or NA (Not Available/Not Applicable).
- **Year Released:** The four-digit year when the material was released for commercial use or NA (Not Available/Not Applicable).
- **Hybrid Designation:** Hybrid names are preceded with the Punycode "×" multiplication character to improve database searching. However, caution is urged since the exact hybrid lineage has not been scientifically verified for some of the earlier data. This column lists an applicable hybrid name that may associate with the listed parental species name entries.
- Many column entries only allow specific character set entries using SQL check constraints to help maintain data entry integrity. Examples include the "Sex" (single character), "Category" (variable characters), and "Year selected" (4 digits). Some fields also allow "NA" values that designate "Not Available/Not Applicable" data.
- The Canadian database records have a flat single table appearance similar to a database view making it possible to display them in a single spreadsheet worksheet. This single table view process makes it easier for users to understand and view the data simply using the various search fields to access the entire database. Searching the database is case insensitive and does not require the exact case of the intended values. Below are sample searches that help describe the data:
- Searching the Clone Directory Database [6] for the clone "Name", "a69" can be entered as, "A69" or "a69" both returning 65 results.
- The aforementioned search returns "AK50" as a "Name" column value without a space, while other "AK" entries have spaces.
- Searching the "Name" fields for "ak" or "AK" both return 39 results including "AK50".
- Searching the "Name" fields for "ak 30" and the "Female Parent Clone" column for "a69" returns two "AK 30" "Name" records. This is possible because the "ID" fields are different. Also, note that the "Current Status" and "Data Source" values are different and may explain the discrepancy.
- Searching the "Female Parent Clone" column for "473-5070" returns ten records with the "Female Parent Clone" value of: "473-5070; FR17; Fraser River South-BC, Canada".

However, searching the "Name" column for "473-5070" or "FR17" returns no records. The database search tool may not be able to identify parents also included in the "Name" clone column.

*1.3. Characteristics of the Open4st Database Model*

The Open4st database [7] was developed in March 2011 to provide online access to the Open4st project clones, families, and related experimental data [8]. It uses the open-source PostgreSQL relational database [9] and the online DBKiss database application [10] to allow read-only access to the database tables and views. It contains an SQL editor that allows custom SQL queries for more in-depth custom reports and can save previous queries for later retrieval. This database process is designed to be used as a central repository by importing data via CSV files, creating SQL queries for specific reports and views for "big picture" cumulative annual summaries that can be exported back to CSV files for R Programming and further user analysis.

An open-source development copy of the Open4st database [11] (aka "r4st") is available via the pmcgover/24dev-demo GitHub public repository. It is a prototyping add-on process for the OSGeo Live DVD [12] that allows users to review, modify, and execute the open-source code using the MIT license [13]. To access this material, users can review the GitHub 24dev-demo documentation, download the latest 24dev-demo release, and install it on an OSGeoLive system [12].

The r4st database documentation provides a high-level description of the 24dev-demo process. The r4st/csv folder contains the CSV files that are loaded into the database with scripts from the r4st/bin folder to create the tables and views. This is essentially a build process that drops and recreates all of the existing data each time the scripts are activated. This allows for easy modifications but could be deactivated and used long term without the build process. The final build step creates a single database dump file of the entire database that can be copied to different server environments and optionally configured from the default write access to read-only access. Key Open4st tables include the Plant (epithet details), Family (parent details), Pedigree (parent/child lineage details), test_detail (annual nursery details), taxa (clone or family species details), and field_trial (field trial details) tables. Related data are associated between tables using foreign key relationships.

Figure 1 shows the current online Open4st DBKiss database application that contains 11 tables and 40 views. Below are key usage notes for the DBKiss database application [10] used throughout this POC.

- Database Tables are displayed on the left side and views on the right. Clicking on a table or a view displays its contents similar to a spreadsheet. Database tables in this context are based on actual spreadsheet data that are loaded into the database.
- Database views can display data from different tables in a variety of reports and summaries with different levels of aggregation.
- Clicking on the view "Count" header displays the count of each view.
- Below are navigation options for the tables or views (Figure 2: Open4st Plant Table Listing):
  - Selecting the "full content" box and clicking the "Search" button expands rows to include all data.
  - Export any given page by clicking one of the "Export to CSV:" options.
  - Click on any column header to reverse the entire column sort order.
  - The user can search any column by entering text in the search box below the "All tables" link, then select the related column from the next drop-down list. Searches are case insensitive.
- The Execute SQL pages allow the user to enter SQL Queries to further refine the search. This is an advanced feature.

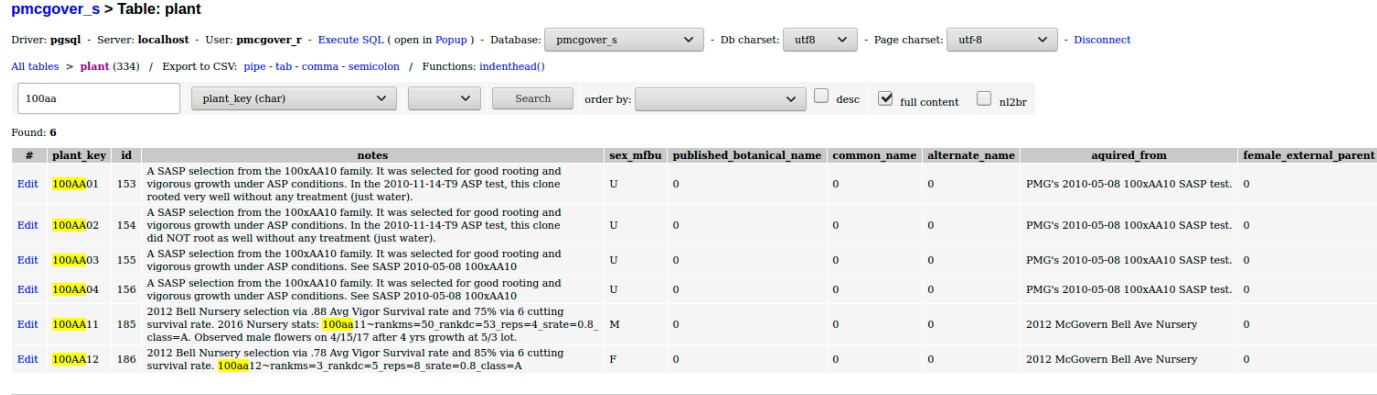

**Figure 1.** The Open4st Online Database: Table and View Listing.

**Figure 2.** The Open4st Online Database: Plant Table Listing.

The Open4st database was developed for a restricted set of users. It does not have time-tested exposure or follow column naming standards similar to the IPC *Salix* or Canadian databases. The species data is derived from a "Taxa" table, which does not follow consistent binomial or species naming conventions.

### 1.4. Characteristics of the Integrated IPC Salix Cultivar Database Proof-of-Concept

Each database model has pros and cons for its respective processes. Given these contrasting models, the intent of this POC is to demonstrate an SQL database to store existing IPC *Salix* cultivar information and provide users with a format to compare and submit new *Salix* cultivar entries. An integrated approach from all three models is proposed using three tables: Epithet, Family, and Pedigree.

The database tables can be displayed online or generated from a desktop application with desired views copied to an online spreadsheet. Users can view the flat inline Epithet and Family records to understand relationships and allow them to submit their new *Salix* cultivar epithet and family entries. A potential scenario could involve users submitting their Checklist entries to the *Salix* administrator for review; the entries are then entered into a test environment by IT resources for final review and promotion to a production online environment.

### 1.5. Integrated IPC Salix Cultivar Database Use Case

The following use case scenario describes the interactions, events, and flow steps between the actors (participants) and the various systems (Figure 3).

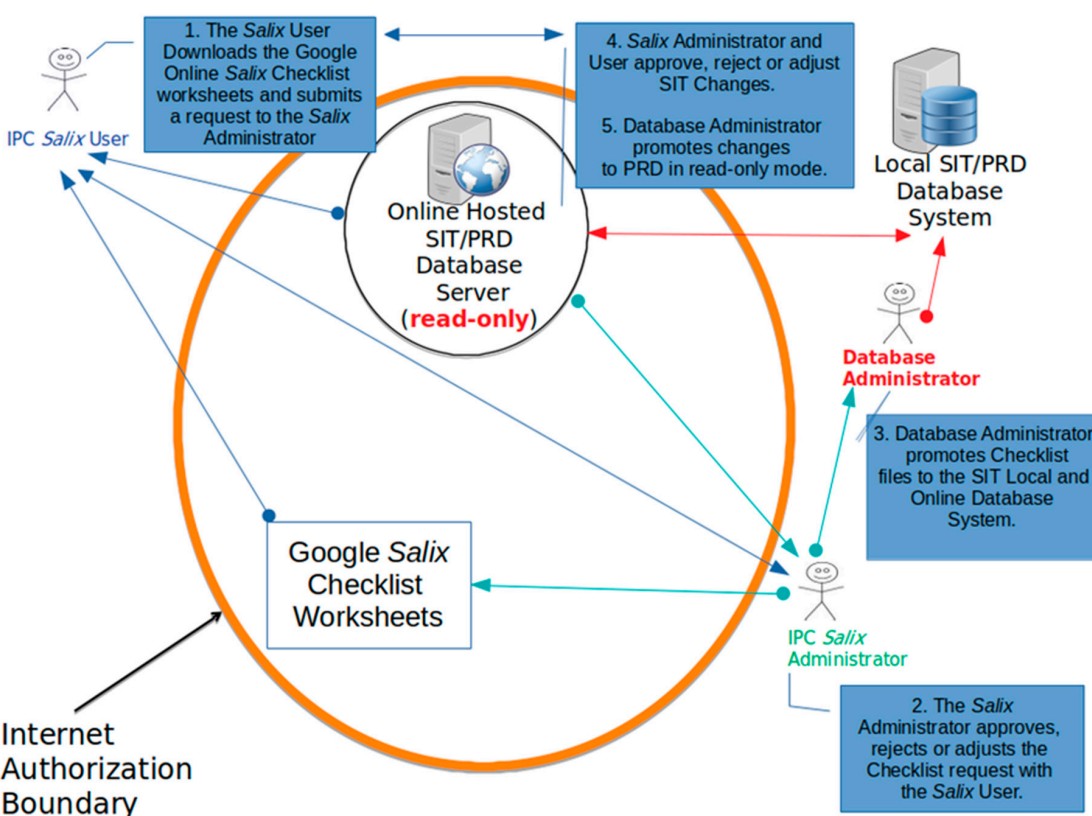

**Figure 3.** *Salix* Checklist use case flow diagram.

### 1.5.1. Brief Description

This use case describes a local and online database system to store IPC *Salix* Cultivar information populated by a user Checklist registration process.

### 1.5.2. Actors

1.  IPC *Salix* Administrator: (e.g., Yulia Kuzovkina).
2.  IPC *Salix* User: A user submitting an IPC *Salix* Checklist request that should be approved by Kuzovkina before it is uploaded to the database.
3.  Database Administrator (DBA): (e.g., Patrick McGovern).
4.  Local Database System: Located on the DBA's local desktop computer with production (PRD) and pre-production (SIT or Staging) database environments.
5.  Online Database System: Located on an online hosted database server with SIT and PRD database environments.
6.  Online *Salix* Checklist Views: The checklist views vw3_checklist_root_level_epithet [14] and vw4_checklist_epithet_family [15], stored on the Online and Local Database System.
7.  Online *Salix* Checklist Worksheets: The user-accessible Google worksheets with the Online *Salix* Checklist View data that is documented and formatted to allow users to copy and submit their own root-level epithet and epithet/family level records. See Google worksheets: vw3_checklist_root_level_epithet [16] vw4_checklist_epithet_family [17] and a CollaboratedChecklist [18] for all parties to collaborate on the proposed Checklist entries.

### 1.5.3. Triggers

The system is triggered when the IPC *Salix* User submits their IPC *Salix* Cultivar files to the IPC *Salix* Administrator for approval.

### 1.5.4. Pre-Conditions

1.  The Local and Online Database Systems have pre-populated table and view data (e.g., Epithet, Family and Pedigree tables, and Checklist views).
2.  The Online Salix Cultivar Worksheets (vw3_checklist_root_level_epithet [16] and vw4_checklist_epithet_family [17]) have the latest version of the vw3_checklist_root_level_epithet [14] and vw4_checklist_epithet_family [15] data from the Online Database System and are available for IPC Salix user access.

### 1.5.5. Basic Flow of Events

1.  The IPC *Salix* Users download the Google Online *Salix* Cultivar Worksheet with the vw3_checklist_root_level_epithet and vw4_checklist_epithet_family view data and submit their new root epithet and/or family epithet *Salix* record request to the IPC *Salix* Administrator.
2.  The IPC *Salix* Administrator reviews the IPC *Salix* user Cultivar submission and communicates with the user to approve, reject, or adjust the request.
3.  The approved Cultivar files are entered by the DBA into the SIT Local Database System, which is then copied to the SIT read-only Online Database System viewable by all users.
4.  The IPC *Salix* Administrator and user approve, reject, or adjust the SIT changes.
5.  The approved changes are then promoted by the DBA and copied to the PRD Local and Online read-only Database systems.

### 1.5.6. Special Requirements

1.  The complexity of the system may likely require dedicated IT personnel to implement and maintain this process long term.
2.  Any other system changes are also tested first in SIT, and then promoted to PRD. Version control for database and/or Checklist file updates are considered.
3.  The Local Database System can be tested locally via the 24dev-demo [11] add-on with the OSGeo-Live DVD [12], USB flash drive [19], or the VirtualBox application.

*1.6. Technical Information*

Additional methodologies, technical details, and screencast videos are also available [20].

**Author Contributions:** Conceptualization, methodology, software, formal analysis, and writing—original draft preparation, P.N.M.; validation, writing—review and editing, Y.A.K. and R.Y.S. All authors have read and agreed to the published version of the manuscript.

**Funding:** This research received no external funding.

**Data Availability Statement:** The data presented in this study can be available on request from the corresponding author.

**Acknowledgments:** We thank Ron Zalesny, Jeffery Jackson, Bradford Bender, and Harvey Anderson for their contributions to this project. Special thanks to the Canadian Forest Service for sharing their database, plant materials, and expertise over the years. The authors also thank the Poplar and Willow Council of Canada for providing access to their clone directory database and Samir Patel for his editorial advice and MDPI formatting expertise.

**Conflicts of Interest:** The authors declare no conflict of interest.

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
