# Peer review of "Short Communication: IPC Salix Cultivar Database Proof-of-Concept"

_forests, doi:10.3390/f12050631_

Round 1
Reviewer 1 Report
The manuscript does not have the character of a research article. It is just a description of starting the database.
Basic (general) shortcomings:
1. The goal of the work is not defined in the manuscript.
2. There is no research question/hypothesis in the manuscript.
3. The manuscript has no chapter Conclusion.
4. The Discussion chapter exists, but its content is not Discussion. There is no critical analysis of the solution. Complete lack of comparison with similar solutions (and there are many similar solutions for biological data).
Article from the IT point of view:
The text is an inappropriate mixture of intent description text, installation instructions and manual.
The authors used common IT tools, but no longer justified the choice or possible alternatives
They used PostgreSQL as a data repository - in what version, in what mode? Missing criteria for selecting this DBMS.
The figures (1, 2, 4) show a view of the data only using the free DBKiss application. From this, I conclude that the authors have not developed any custom application interface.
According to the Figures (1, 2, 4) database runs only in localhost mode. The authors did not try to deploy the solution to the production server (?). Therefore, there can be no data from performance and availability testing.
If the solution is to serve more users, it is necessary to define suitable DB roles. There is no information about this in the text (gross shortcoming from the point of view of DB management).
The "sql_query" is repeated several times in the text. - What SQL standard is implemented (SQL-2, SQL-3, SQL-MM)? To what extent is the standard implemented?
Completely unsuitable structure chap.1.5. A chapter in the range of 1 line is not allowed (eg 1.5.1). The numerical designation of bullet points is mixed with the numbering of chapters.
Chapter 2.1.2 does not belong at all to the scientific text.
This is no longer technical documentation, this is a guide for beginners on how to set up DB.
The paragraph on running the phpinfo file is from the beginner's manual, as well as the other steps of putting the usual triad into operation (webserver, database and scripting language).
What is the real result and benefit?
Unfortunately, there is no novelty or research in the manuscript.
The research could focus on (eg) testing the distributed DB mode, testing data synchronization, methods of generating a global ID for individual records, developing an interface for semantic querying using SPARQL ...
Reviewer 2 Report
The paper presents both contemporary and long-lasting challenges of the collection, nomenclature, collection of databases of plant species, created material in form of varieties, cultivars and clones both from the perspective of botanic nomenclature and from the perspective of their commercial use.
Challenges related to the unification of nomenclature in forestry and also in the gathering of common list of species and cultivars presents long term problem for which this manuscript presents one of the possible solutions. These solutions not only reflect in the databases construction itself, but also in the process of the database inputs, update and corrections that are made through the stepwise process of the approval from all the participants in the use of the database.
The paper presents well description and comparison of different plant databases emphasizing their advantages and disadvantages. This would present good information both for researchers and plant producers and also for researchers and programmers that are dealing with databases. Therefore, the paper is definitely worth publishing in the Forests.
Since the form and concept of the manuscript does not align with the classic research paper, my suggestion is that the manuscript should exclude Materials and Methods, Results and Discussion chapters and to be presented in its current form as either short communication or mini review paper.
